# COVID-19 Vaccine Coverage and Sociodemographic, Behavioural and Housing Factors Associated with Vaccination among People Experiencing Homelessness in Toronto, Canada: A Cross-Sectional Study

**DOI:** 10.3390/vaccines10081245

**Published:** 2022-08-03

**Authors:** Lucie Richard, Michael Liu, Jesse I. R. Jenkinson, Rosane Nisenbaum, Michael Brown, Cheryl Pedersen, Stephen W. Hwang

**Affiliations:** 1MAP Centre for Urban Health Solutions, St. Michael’s Hospital, Toronto, ON M5B 1W8, Canada; mliu02@hms.harvard.edu (M.L.); jesse.jenkinson@unityhealth.to (J.I.R.J.); rosane.nisenbaum@unityhealth.to (R.N.); cheryl.pedersen@unityhealth.to (C.P.); stephen.hwang@unityhealth.to (S.W.H.); 2Harvard Medical School, Boston, MA 02115, USA; 3Dalla Lana School of Public Health, University of Toronto, Toronto, ON M5S 1A4, Canada; stormbringer.brown@mail.utoronto.ca; 4Department of General Internal Medicine, University of Toronto, Toronto, ON M5S 1A4, Canada

**Keywords:** COVID-19 vaccines, homelessness, public health, Toronto, Canada

## Abstract

People experiencing homelessness were prioritized for COVID-19 vaccination in Toronto, Canada, due to the high risk of infection and associated complications relative to the general population. We aimed to ascertain COVID-19 vaccine coverage in this population and explore factors associated with the receipt of at least one dose. We collected survey and blood sample data from individuals ages 16+ recruited by random selection at 62 shelters, hotels and encampment sites between 16 June 2021 and 9 September 2021. We report vaccine coverage by dose number and explored sociodemographic, behavioral, health and housing factors associated with vaccination using multivariable modified Poisson regression. In total, 80.4% (95% CI 77.3–83.1%) received at least one vaccine dose, and 63.6% (CI 60.0–67.0%) received two or more doses. Vaccination was positively associated with age (every 10 years adjusted rate ratio (aRR) 1.05 [95% CI 1.03–1.08]), and receipt of influenza vaccination (aRR 1.19 [95% CI 1.11–1.27]). Factors negatively associated with vaccination included female gender (aRR 0.92 [95% CI 0.85–1.0]), Black racial self-identification (aRR 0.89 [95% CI 0.80–0.99]) and low frequencies of masking in public places (aRR 0.83 [95% CI 0.72–0.95]). COVID-19 vaccine coverage is very high among people experiencing homelessness in Toronto, suggesting advocacy and outreach efforts may have been effective.

## 1. Introduction

Homelessness, where an individual lacks stable, permanent and appropriate housing, places individuals at greater risk for contracting severe acute respiratory syndrome coronavirus 2 (SARS-CoV-2) [1,2]. Due to disproportionate physical, mental and social burdens that increase morbidity relative to the general population [3,4,5], people experiencing homelessness have also been found to be at heightened risk of health complications following COVID-19 infection [1,6]. In recognition of the excess risk faced by the estimated 235,000 Canadians (about 0.7% of the population) who experience homelessness each year [7], the province of Ontario, Canada heeded calls in February 2021 to revise its vaccination plan to include people experiencing homelessness in Phase 1 (representing the highest priority level available) [8,9].

Since then, COVID-19 vaccines were made available to people experiencing homelessness in Toronto, Canada, where at least 18,096 [10] persons experiencing homelessness resided in 2021. Toronto, on Treaty 13 territory, is the most populous city in Ontario, a jurisdiction that provides universal health care through a single payer system; as such, COVID-19 vaccines were free to all who reside in the province, regardless of citizenship status or documentation [11]. People experiencing homelessness were encouraged to receive vaccines at mass-vaccination centers; in addition, recognizing the additional barriers faced by this population in receiving vaccination [12,13], local public health authorities in collaboration with community agencies supplemented mass vaccination clinics with outreach clinics at emergency shelters, COVID-19 physical distancing shelter hotels (Hotels temporarily commissioned as shelters for physical distancing), encampments and drop-in centres [14,15,16,17,18,19]. To further mitigate against expectations of low uptake, Peer Ambassadors (later called Peer Champions) were further deployed in support of vaccine clinics to engage with people having vaccine-related questions or concerns [14].

Early evidence indicates COVID-19 vaccine coverage among people experiencing homelessness is low, especially relative to the general population. In six US public health jurisdictions, vaccine coverage among people experiencing homelessness was reported to be between 10.5% and 37.9% lower (1+ dose) relative to the general population by August 2021 [20]. In Minnesota, people experiencing homelessness had 33.7% coverage compared to 64% coverage in the general population, by November 2021 [21]. Among US veterans, people experiencing homelessness had 45.8% coverage (1+ dose) compared to 64.3% coverage in the general population by September 2021 [22]. In Ontario, Canada, people with a recent history of homelessness had a vaccine coverage (1+ dose) rate of 61.4%, about 25% lower than the general population [23]. In Denmark, people under 65 years of age experiencing homelessness had the lowest coverage rate (54.6%) among groups studied by October 2021, at an incidence rate ratio of 0.5 (males) and 0.4 (females) compared to the general population [24]. In Italy, vaccination coverage in homeless settlements ranged between 13% and 36% from June to September 2021, compared to 79% in the general population [25]. Barriers to access, competing priorities and mistrust of government institutions are all believed to contribute to low vaccination uptake in this population [25,26,27,28,29], as is COVID-19 vaccine acceptability [26,27,28,29,30]. Current reports suggest that trusted healthcare providers may be key to enhancing vaccination confidence and overcoming hesitancy [22,23]. Very few studies report vaccination-related attitudes by housing status (sheltered vs. unsheltered), but one report suggests this status is unimportant [29].

Although the City of Toronto reported in January 2022 that 76% of emergency shelter users received one dose and 65% received two doses [31], COVID-19 vaccine coverage and factors associated with vaccination among people experiencing homelessness in Toronto are not well understood. In the present study, we harness the baseline survey and biological sample results of the *Ku-gaa-gii pimitizi-win* cohort study (formerly known as the COVENANT study) to report COVID-19 vaccination coverage and factors associated with vaccination among a random sample of people experiencing homelessness in Toronto, Canada.

## 2. Materials and Methods

### 2.1. Setting and Design

This cross-sectional analysis uses data collected between June and September 2021 from participants of the *Ku-gaa-gii pimitizi-win* study, a prospective cohort study of people experiencing homelessness in Toronto, a city on Treaty 13 territory in Ontario, Canada. *Ku-gaa-gii pimitizi-win*, which translates in English to *life is always/forever moving*, is a spirit name given in ceremony by Elder Dylan Courchene from Anishnawbe Health Toronto. This name reflects and honours the movement of homeless individuals across the land, the spirit and growth of the land we are on, and the force that connects us all to the future. The *Ku-gaa-gii pimitizi-win* study protocol is published elsewhere [32]. In this region, COVID-19 vaccines are available at no cost to residents and were approved by Health Canada on 9 December 2020 (Pfizer [New York, United States]) [33], 23 December 2020 (Moderna [Cambridge, United States]) [34], 26 February 2021 (AstraZeneca [Cambridge, United Kingdom]) [35] and 5 March 2021 (Johnson & Johnson [New Brunswick, United States]) [36], respectively. The government of Ontario, which is responsible for vaccination strategy in the province, prioritized vaccination for Ontarians according to their risk level [9], with people experiencing homelessness receiving Phase 1 (highest) priority beginning February 2021 [8]. However, because vaccine demand exceeded availability until at least early summer 2021, the wait time was lengthened to four months between doses in order to allow more Ontarians to receive protection from a first dose [37]. 

This study follows the Strengthening the Reporting of Observational Studies in Epidemiology (STROBE) guidelines (Appendix A) and received ethics approval (REB# 20-272) from the Research Ethics Board at St. Michael’s Hospital, Unity Health Toronto. 

### 2.2. Participants and Recruitment

The reference population included approximately 6000 individuals then-residing in any of 61 participating emergency shelters and COVID-19 physical distancing shelter hotels in the City of Toronto, as well as residents in one urban encampment site. To recruit a minimum of 670 participants, at each recruitment site (save the encampment, where we approached everyone available due to logistical constraints) a random number schedule was assigned to beds or rooms to select potential participants. A sampling fraction was enforced to ensure each site contributed an appropriate number of participants. Selected potential participants, if available, were then provided detailed information about the study and asked to consent to participate. To be eligible, potential participants had to be 16 years of age or older, be willing to participate in follow-up intervals, and be able to provide informed consent. If informed consent was provided and the individual was eligible, participants then provided biological samples and were interviewed on the spot.

### 2.3. Characteristics of Participants

Participants completed a detailed baseline survey, which solicited self-reported sociodemographic information (age and self-identified gender, race, Indigenous identity, citizenship status, level of education, paid work since start of the pandemic), chronic health conditions diagnosed by a physician (such as hypertension, diabetes, asthma), relevant health-related behaviours such as receipt of influenza vaccination, self-reported history of COVID-19 infection and level of observance of public health recommendations endorsed in Ontario at the time (such as masking in public places and practicing physical distancing). Participants also provided a recent (90-day) housing history, following the residential time-line follow-back inventory method [38]. Survey questions were previously used in large-scale studies of homeless persons [39], recommended by the Canadian Institute for Health Information [40] or adapted from CDC and Statistics Canada COVID-19 related surveys [41,42]. The survey instrument (available as a Supplement with the study protocol [24]) was further reviewed with community partners and piloted with individuals having lived experience to ensure questions did not cause unnecessary discomfort or harm.

Participants further provided a saliva sample (Swish and gargle method collected in a Leakbuster container), which was tested using standard methods [43] to detect current infection (RT-qPCR). They also provided a blood sample (either in a plasma tube [BD365985] or as a dried blood spot [Whatman 903]) which was tested with highly sensitive and specific chemiluminescence-based enzyme-linked immunosorbent assays (ELISA) [44] to help ascertain the presence of current or past COVID-19 infection or vaccination-induced antibodies (spike protein trimer, spike protein receptor-binding protein, and nucleocapsid antigen).

A full list of characteristics and variable definitions used in this study is detailed in Appendix A. 

### 2.4. Outcome Measures

Our primary outcome of interest was receipt of at least one dose of any Health Canada approved COVID-19 vaccine product at or before the time of the interview date. We ascertained vaccination status through a combination of self-report data and serological sample results; final determination of vaccination status was adjudicated as described elsewhere [32]. Briefly, vaccination status and date of vaccination were accepted as-given, unless the date was outside the range of possible vaccination dates in Ontario (before 14 December 2020), in which case the vaccination event was accepted to have occurred but the date was rendered missing. Participants were categorized by vaccination status (vaccinated with at least one dose; or unvaccinated).

### 2.5. Statistical Analysis

We compared sociodemographic, behavioural, health and housing characteristics of participants by vaccination status as of their interview date. Characteristics were summarized as counts and proportions, means and standard deviations (SD), or median and interquartile range (IQR), with the significance of difference assessed using chi square, *t*-test, one-way ANOVA, Wilcoxon rank-sum and Kruskal–Wallis tests, as appropriate. We calculated 95% confidence intervals (CI) for overall coverage rate using the Wilson Score method for proportions [45].

The cumulative incidence of COVID-19 vaccination was calculated from 14 December 2020 (first date of vaccination availability in Ontario) to 9 September 2021. Coverage rates over time were measured as the number of people in the cohort who were vaccinated and whose vaccination date is known divided by the number of individuals in the cohort at risk. Participants unvaccinated at the date of interview were censored after their interview date. For descriptive purposes, we also plotted vaccine coverage rates for the adult (18+) population of Toronto for the same period, using publically available vaccination data [46] and the 2021 census population estimate [47].

As our outcome (vaccination) occurred at a high rate, we applied a modified multivariable Poisson regression with robust error variance using a log link to estimate the adjusted incidence rate ratio (aRR) and 95% CI of receiving one or more doses of any COVID-19 vaccine as of the baseline interview date. Factors associated with vaccination in existing literature or with significant differences in the bivariate tests were considered for inclusion in the model. Correlation coefficients among variables that could suggest multi-collinearity were estimated and assessed prior to modelling. For variables with missing data, multiple imputations were used to generate a minimum number of imputed datasets (in this case, five), the regression results of which were pooled according to Rubin’s rules [48] using PROC MIANALYZE. 

We conducted all analyses using the SAS enterprise guide v 7.1. Throughout, statistical tests were 2 tailed and *p*-values of less than 0.05 were considered statistically significant.

## 3. Results

We recruited from 2643 randomly selected, assigned beds from 61 shelters, COVID-19 physical distancing shelter hotels and one encampment site across Toronto (Figure 1). Of these, 1098 (or 41.5%) and 443 (or 16.8%) individuals were unavailable or not present to be recruited, respectively. Of the 1102 individuals approached, 12 were deemed unable to consent and 354 individuals (or 32.1%) refused to participate. We thus recruited a total of 736 unique participants, representing a 66.8% recruitment rate among individuals approached and a 27.8% recruitment rate overall. Of these 736 participants, 728 had useable serological data (i.e., sufficient sample quantity and quality to produce test results) for this analysis.

Table 1 reports the characteristics of the cohort. Participants were predominantly male (66.1%) and the mean age was 46 years (SD 14.7 years). A little over one in ten (10.3%) self-identified as Indigenous. Approximately half (48.5%) self-identified as White, 21.8% self-identified as Black and 18.5% self-identified as other/multiple racial categories. More than one in ten (12.4%) were permanent residents, 7.6% were refugee claimants, and 3.7% had temporary (e.g., visitor, student) or other non-official status. Over a third (36.0%) had completed post-secondary (i.e., university, college, vocational/technical school or professional school) education. Nearly one in four (22.7%) engaged in paid work since March 2020. Most participants stayed primarily in shelters (39.4%) or COVID-19 physical distancing shelter hotels (45.1%) over the past 90 days immediately preceding the interview. Nearly half of the participants (48.1%) had at least one diagnosed chronic condition. Nearly 30% (29.8%) received an influenza vaccine in the past season. More than four out of five participants reported often or always following each listed public health guideline.

Of these participants, 585 individuals, or 80.4% (95% CI: 77.3–83.1%) of the cohort, received at least one COVID-19 vaccine dose, and 463 individuals, or 63.6% (95% CI: 60.0 –67.0%), had at least two doses. Vaccine product received for each dose is summarized in Appendix A; briefly, the vast majority of vaccines received were Pfizer (approximately 50% of first and second doses) or Moderna (approximately 30% of first and second doses) products, the two first vaccines approved by the Canadian Government. While the recommended spacing between doses varied over the observation period, a conservative spacing of 8 weeks between doses (well within the mandated four months [37] when most vaccinations occurred) would result in most participants with one dose as of the survey date (77 of 122) being ineligible for their second dose. Figure 2 displays the cumulative incidence (with 95% CI) of vaccination by the timing of vaccination among those able to report their vaccination date, in comparison to the general population of Toronto. Though our cohort initially had greater uptake, rates of vaccination among the general population of Toronto quickly caught up, by early May 2021 (dose 1) and the middle of June 2021 (dose 2). 

Table 1 also presents characteristics of participants by vaccination status (at least one dose vs. unvaccinated). Characteristics significantly associated with receipt of vaccination include: older age (vaccinated mean age 47.8 years vs. unvaccinated mean age 39.1 years); self-identification as White (48.4% vs. 35.0%); staying mostly in shelters (40.7% vs. 34.3%) or COVID-19 physical distancing shelter hotels (45.8% vs. 42.0%) in the past 90 days; having at least one chronic condition (50.1% vs. 39.9%); having received the seasonal influenza vaccine (34.9% vs. 9.1%); or reporting high levels of masking in public (88.2% vs. 78.6%). Conversely, characteristics significantly associated with not receiving vaccination include: self-reporting as Black (18.3% vaccinated vs. 32.9% unvaccinated); having high school or secondary level education only (31.8% vs. 45.5%); engaging in paid work since the start of the pandemic (20.7% vs. 30.8%); or reporting low levels of masking in public places (11.1% vs. 20.6%). 

Characteristics associated with greater representation among zero, one and two-dose vaccination status were very similar to that of 1+ dose vaccination status, and are available in Appendix A.

In pooled results from the multivariable analysis (Table 2), age remained positively associated with the receipt of one or more vaccine doses (every 10 years: aRR 1.05 [95% CI 1.03–1.08]). Individuals who received an influenza vaccine in the past season had higher vaccination rates than those who did not receive an influenza vaccine (aRR 1.19 [95% CI 1.11–1.27]). Conversely, lower rates of vaccination were found among people identifying as female (aRR 0.92 [95% CI 0.85–1.0]), Black (aRR 0.89 [95% CI 0.80–0.99]), or who reported low levels of masking in public places (aRR 0.83 [95% CI 0.72–0.95]).

## 4. Discussion

We found that 80.4% of our cohort received at least one COVID-19 vaccine dose by their interview, occurring between June and September 2021. 63.6% had received two or more doses. Most participants with one dose were ineligible for their second dose as of their survey date. We also found that vaccination in our cohort was positively associated with increasing age and receipt of influenza vaccine during the past season, and negatively associated with female gender, self-identification as Black and lower levels of masking in public places. Primary housing type within 90-days of the baseline interview was not associated with vaccination status after adjustment in multivariable models.

Factors we found to be significantly associated with COVID-19 vaccination are congruent with existing literature on COVID-19 vaccination uptake or hesitancy in this population [21,22,23,24,25,26,27,28,29,30,49]. Older age, a strong predictor of adverse COVID-19 related outcomes, is often positively associated with greater vaccination uptake [13,14,15,16,22], as is influenza vaccination [21,22,23,29,30]. Individuals self-reporting as Black are consistently found to have lower vaccination uptake or greater hesitancy, which is likely a consequence of structural racism producing deep mistrust of governments and/or health authorities, which can also hinder access to healthcare [21,22,28,29,30,49,50]. Associations by gender or sex are less consistently reported, but our findings are consistent with most reports that indicate female gender is associated with greater vaccine hesitancy or uptake [24,26,49]. Being sheltered as opposed to unsheltered was not associated with vaccination in our analysis; this is consistent with a previous study among people experiencing homelessness in Los Angeles [29]. Finally, we are, to our knowledge, the first to report a positive association between masking in public places and vaccination. Masking in public places was at this time mandatory in Ontario, and in this setting, masking might have been a proxy for trust in or willingness to comply with public health recommendations (such as vaccination) more generally [29].

The coverage rate found in this study differs from others reported to date [20,21,22,23,24,25,26,27,28,29,30,31] in that we found very high vaccine coverage among people experiencing homelessness in Toronto. In our report, first-dose coverage by summer 2021 was over 80%, which is higher than the rates reported by the City of Toronto four months later (76%) [31], or in a recent Ontario-based report by September 2021 (61.4%) [23]. This coverage rate was also substantially higher than what was reported in the US (from 10.5% to 37.9% by August 2021 [20]; 33.7% by November 2021 in Minnesota [21]; and 45.8% by September 2021 among veterans [22]), Denmark (54.6% by October 2021 [24]); and Italy (between 13% and 36% between June and September 2021 [25]). As most of the other studied settings had general populations with rates of vaccination similar to that of Canada [51], it is not possible to discount the difference only as a matter of regional differences in vaccine acceptability. Indeed, most reports, including the Ontario report which covers our region [23], indicate substantial gaps in uptake among people experiencing homelessness compared to the general population. By contrast, first-dose coverage in our report approximates that of the general adult population of Toronto of the same period (74.8% on 16 June, increasing to 84.3% as of 9 September) [46]. 

There are a few potential factors explaining this result. First, in Toronto the vaccination campaign was aggressive, and in particular, took many measures to overcome vaccination-related barriers faced by people who experience homelessness. Effective advocacy led to this population receiving the highest priority both in the provincial rollout of COVID-19 vaccines [8] and in the City of Toronto’s vaccination program [15]; following this, numerous local agencies who interact regularly with people experiencing homelessness outside of the COVID-19 vaccination effort (such as Community Health Centres and service providers like Anishnawbe Health Toronto and Inner City Health Associates) were integrated in the vaccination effort in deploying targeted mobile clinics to emergency shelters, COVID-19 physical distancing shelter hotels, encampments and other sites [16,17,18,19]. By May 2021, every emergency shelter and COVID-19 physical distancing shelter hotel site in Toronto received at least one such outreach clinic [52]. Further, the deployment of peer ambassadors to outreach clinics provided additional trusted sources for vaccine-related information and discussion [14]. This strategy may have greatly increased information sharing from trusted parties, which has previously been shown to increase vaccine confidence [12,13].

Second, while our study design likely resulted in more generalizable results than would be possible from administrative data focused on specific groups [22] or which required specific types of contact (e.g., healthcare, shelters) [21,22,23,24], our data were obtained from individuals who consented to participate, and it is plausible that these individuals may be more likely to accept COVID-19 vaccination than individuals who declined to participate. Such a sampling process could result in a selection bias towards overestimating vaccination coverage rates among people experiencing homelessness. In addition, the majority of recruitment for the *Ku-gaa-gii pimitizi-win* study occurred in shelters and COVID-19 physical distancing shelter hotels. To the extent that participants remained in these settings for long periods of time, they would have received significant outreach in comparison to individuals sleeping rough or in encampments, and those generally avoidant of locations targeted by outreach [12,13,17]. Our own data suggests individuals who stayed primarily or exclusively in shelters or COVID-19 physical distancing shelter hotels in the past 90 days were more likely to be vaccinated (though this became non-significant after adjustment). Approximately 90% of people experiencing homelessness in Toronto are sheltered [53]; nevertheless, a sample that more systematically included individuals sleeping rough or in encampments throughout the vaccination period may have yielded a lower coverage rate.

### Limitations and Future Research Needs

While we recruited randomly across a large number of emergency housing sites and from one encampment, we were unable to sample from the street or from other vulnerable housing settings (e.g., people staying temporarily with friends or family); thus, our results should only be generalized to people experiencing homelessness in emergency housing settings. Further, approached individuals could opt to not participate. As such, our sample potentially exhibits some selection bias in favour of people having sufficient time and/or willingness to participate in research studies. Finally, much of our information relies on self-report, which can become unreliable over time and/or suffer from social desirability bias, particularly in vulnerable populations facing significant stigma [54]. 

While we provide important information about COVID-19 vaccination coverage in Toronto and factors associated with being vaccinated, our results do not permit us to further understand the reasons why certain people choose to accept or reject COVID-19 vaccination. An on-going *Ku-gaa-gii pimitizi-win* qualitative sub-study will at a later date report on this topic. Future work should also more fulsomely investigate coverage in encampments or among people sleeping rough, as well as the impact, if any, of vaccination on protecting people experiencing homelessness from COVID-19 infection and related complications.

## 5. Conclusions

In conclusion, our findings suggest that, although people experiencing homelessness face considerable barriers to accessing healthcare services like COVID-19 vaccination, concerted advocacy and outreach can overcome known barriers to vaccination. Individuals residing in Toronto shelters and COVID-19 physical distancing hotels in summer 2021 had very high rates of first-dose vaccination, approximating that of the general adult population of Toronto during the same period. Such levels of vaccination provide vital protection for individuals residing in settings that are especially vulnerable to viral outbreaks [55,56,57]. However, opportunities exist to further improve vaccination coverage among particular subgroups, including unsheltered people experiencing homelessness. Additional public health strategies are needed to understand and address how intersecting experiences of marginalization and oppression facing many of the subgroups identified in our study impact choices and opportunities to receive a vaccination. 

## Figures and Tables

**Figure 1 vaccines-10-01245-f001:**
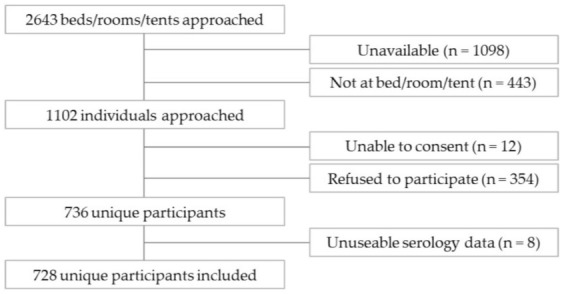
***Ku-gaa-gii pimitizi-win*** recruitment and reasons for non-participation.

**Figure 2 vaccines-10-01245-f002:**
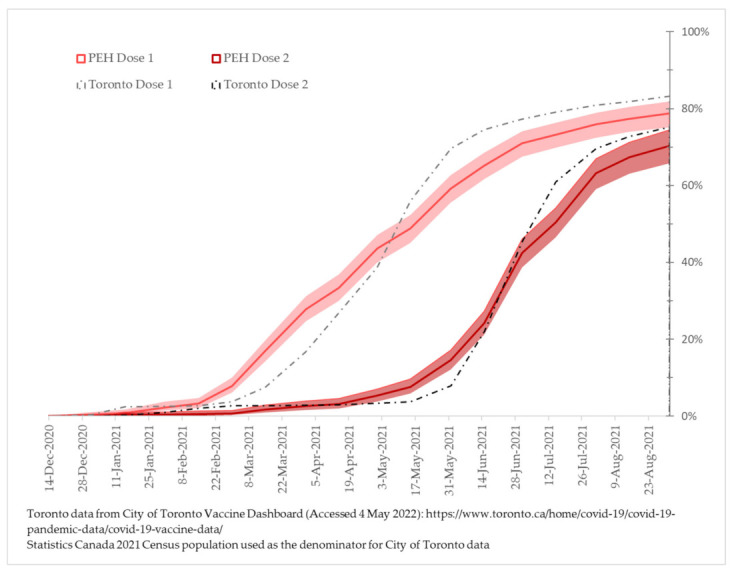
Cumulative incidence (and 95% Confidence Interval) of COVID-19 vaccination by dose number, among (1) *Ku-gaa-gii pimitizi-win* study participants having known vaccination dates, and (2) adult Torontonians overall.

**Table 1 vaccines-10-01245-t001:** ***Ku-gaa-gii pimitizi-win*** participant characteristics at baseline, overall and by vaccination status.

Participant Characteristics		Total (*n* = 728)	No Dose (*n* = 143)	1 + Dose (*n* = 585)	*p*-Value
Age	Mean (SD)	46.1 (14.7)	39.1 (13.8)	47.8 (14.4)	<0.001
	Median (IQR)	46 (34–58)	36 (28–48)	48 (36–58)	<0.001
Self-reported gender	% Male	481 (66.1%)	83 (58.0%)	398 (68.0%)	0.07
	% Female	228 (31.3%)	56 (39.2%)	172 (29.4%)	
	% Other	17 (2.3%)	4 (2.8%)	13 (2.2%)	
	Missing ^1^	2 (0.3%)	0	2 (0.3%)	
Identifies as Indigenous	Yes	75 (10.30%)	12 (8.39%)	63 (10.77%)	0.38
	No	631 (86.68%)	128 (89.51%)	503 (85.98%)	
	Missing ^1^	22 (3.02%)	3 (2.10%)	19 (3.25%)	
Self-reported race	Black	159 (21.84%)	47 (32.87%)	112 (19.15%)	0.002
	Indigenous	27 (3.71%)	5 (3.50%)	22 (3.76%)	
	White	353 (48.49%)	53 (37.06%)	300 (51.28%)	
	Other/Multiracial	156 (21.43%)	31 (21.68%)	125 (21.37%)	
	Missing ^1^	33 (4.53%)	7 (4.90%)	26 (4.44%)	
Citizenship status	Citizen	556 (76.4%)	112 (78.3%)	444 (75.9%)	0.20
	Landed Immigrant	90 (12.4%)	11 (7.7%)	79 (13.5%)	
	Refugee	55 (7.6%)	11 (7.7%)	44 (7.5%)	
	Temporary/Other	20 (2.8%)	6 (4.2%)	14 (2.4%)	
	Missing ^1^	7 (1.0%)	3 (2.1%)	3 (0.7%)	
Highest level of education completed	Less than high school	206 (28.3%)	33 (23.1%)	173 (29.6%)	0.01
	High school	251 (34.5%)	65 (45.5%)	186 (31.8%)	
	Any post-secondary	262 (36.0%)	44 (30.8%)	218 (37.3%)	
	Missing ^1^	9 (1.2%)	1 (0.7%)	8 (1.4%)	
Engaged in paid work since March 2020		165 (22.7%)	44 (30.8%)	121 (20.7%)	0.01
Primary housing type in past 90 days	Homeless shelter	287 (39.4%)	49 (34.3%)	238 (40.7%)	0.009
	Distancing hotel	328 (45.1%)	60 (42.0%)	268 (45.8%)	
	Other setting	113 (15.5%)	34 (23.8%)	79 (13.5%)	
Presence of chronic condition(s) ^2^		350 (48.1%)	57 (39.9%)	293 (50.1%)	0.028
Body mass index category	Underweight/normal	333 (45.7%)	61 (42.7%)	272 (46.5%)	0.46
	Overweight	226 (31.0%)	51 (35.7%)	175 (29.9%)	
	Obese	140 (19.2%)	27 (18.9%)	113 (19.3%)	
	Missing ^1^	29 (4.0%)	4 (2.8%)	25 (4.3%)	
Influenza vaccine in past season		217 (29.8%)	13 (9.1%)	204 (34.9%)	<0.001
Following Public Health Guidelines: wear face mask in public	Low (never/rarely/occasionally)	93 (12.9%)	30 (21.1%)	63 (10.8%)	0.001
	High (often/always)	630 (87.1%)	112 (78.9%)	518 (89.2%)	
	Missing ^1^	5 (0.7%)	1 (0.7%)	4 (0.7%)	
Following Public Health Guidelines: distancing in public places	Low (never/rarely/occasionally)	103 (14.3%)	26 (18.4%)	77 (13.3%)	0.12
	High (often/always)	615 (85.7%)	115 (81.6%)	500 (86.7%)	
	Missing ^1^	10 (1.8%)	2 (1.4%)	8 (1.4%)	
Following Public Health Guidelines: avoid crowded places or gatherings	Low (never/rarely/occasionally)	136 (19.1%)	36 (25.7%)	100 (17.5%)	0.026
	High (often/always)	577 (80.9%)	104 (74.3%)	473 (82.6%)	
	Missing ^1^	15 (2.1%)	3 (2.1%)	12 (2.1%)	
Following Public Health Guidelines: wash hands with soap/sanitizer several times per day	Low (never/rarely/occasionally)	72 (10.0%)	17 (12.0%)	55 (9.5%)	0.37
	High (often/always)	651 (90.0%)	125 (88.0%)	526 (90.5%)	
	Missing ^1^	5 (0.7%)	1 (0.7%)	4 (0.7%)	

^1^ Missing data of any type (refused, do not know, missing) were not included in the denominator when running statistical tests. ^2^ At least one of the following chronic conditions, diagnosed by a physician: Hypertension, diabetes, asthma, chronic lung disease, chronic heart disease, stroke, chronic kidney disease, chronic neurological disorder, liver disease, cancer, HIV/AIDS or an immunological disease other than HIV/AIDS.

**Table 2 vaccines-10-01245-t002:** Factors associated with COVID-19 vaccination (1 or more doses) among *Ku-gaa-gii pimitizi-win* participants with confirmed serology at baseline (June to September 2021) (*n* = 728).

Participant Characteristics		Adjusted Rate Ratio	95% CI	*p*-Value
Age (per ten years)		1.05	1.03–1.08	<0.001
Self-reported gender	Male (ref)	N/A		
	Female	0.92	0.85–1.00	0.040
	Other	1.07	0.82–1.41	0.604
Self-reported race	White (ref)	N/A		
	Black	0.89	0.80–0.99	0.033
	Indigenous	0.99	0.83–1.19	0.930
	Other/Multiracial	1.00	0.92–1.08	0.957
Received influenza vaccine in past season	No (ref)	N/A		
	Yes	1.19	1.11–1.27	<0.001
Frequency of masking in public places	Often/Always (ref)	N/A		
	Never/Rarely/Occasionally	0.83	0.72–0.95	0.009
Primary housing type in past 90 days	Homeless shelter/Physical distancing hotel (ref)	N/A		
	Other settings	0.93	0.82–1.05	0.222

## Data Availability

Due to the vulnerability of the study population and the sensitive nature of the collected data, ethical approval for this study requires that study data remains on secure servers. As such, data presented in this study are not publicly available. However, queries about the data supporting this study can be directed to the Corresponding Author.

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
