# Peer review of "COVID-19 Vaccine Coverage and Sociodemographic, Behavioural and Housing Factors Associated with Vaccination among People Experiencing Homelessness in Toronto, Canada: A Cross-Sectional Study"

_vaccines, 2022, doi:10.3390/vaccines10081245_

Round 1
Reviewer 1 Report
This is a well-prepared manuscript with a high scientific soundness level.
The study aim is important and the methods applied are correct.
Supplementary materials are an important part of the manuscript and provide a comprehensive overview.
Please consider the following minor revisions:
1. Introduction: 1-2 sentences on the proportion of people experiencing homelessness in Canada/Toronto will be informative for the international readers
2. Methods: Lines 106-109 "Participants completed a detailed baseline survey, provided a saliva sample to detect current infection (RT-qPCR), and provided a capillary blood sample (either in a plasma tube or as a dried blood spot) to help ascertain the presence of current or past COVID-19 infection or vaccination-induced antibodies (spike protein trimer, spike protein receptor-binding protein, and nucleocapsid antigen)" Please consider adding 2-3 sentences on laboratory methods and devices/tests used in this study. This basic info may be provided in the supplementary material
3. Please consider adding 2-3 sentences on the limitations of this study, practical implications, and further research needs
Author Response
This is a well-prepared manuscript with a high scientific soundness level.
The study aim is important and the methods applied are correct.
Supplementary materials are an important part of the manuscript and provide a comprehensive overview.
Please consider the following minor revisions:
- Introduction: 1-2 sentences on the proportion of people experiencing homelessness in Canada/Toronto will be informative for the international readers
Response: Thank you for your careful review of our study. While the size and proportion of the population experiencing homelessness in Toronto, as in other locations, is somewhat controversial due to very imperfect enumeration methods, we’ve added the most contemporary estimates for Canada (in the introduction, paragraph 1) as well as for Toronto (in the first section of the Methods, dealing with setting and Design) to help contextualize the extent of this social challenge in our area.
- Methods: Lines 106-109 "Participants completed a detailed baseline survey, provided a saliva sample to detect current infection (RT-qPCR), and provided a capillary blood sample (either in a plasma tube or as a dried blood spot) to help ascertain the presence of current or past COVID-19 infection or vaccination-induced antibodies (spike protein trimer, spike protein receptor-binding protein, and nucleocapsid antigen)" Please consider adding 2-3 sentences on laboratory methods and devices/tests used in this study. This basic info may be provided in the supplementary material
Response: Our protocol provided to some extent this information, but we agree it should be readily available here. We’ve restructured the cited paragraph to include these details.
- Please consider adding 2-3 sentences on the limitations of this study, practical implications, and further research needs
Response: Thank you, we’ve added a few sentences about implications of our results to the Discussion, and created a final section to more explicitly discuss the limitations of the study, as well as future research needs.
Reviewer 2 Report
This study is scientifically correct. The aim of the study is clearly defined. The methodology section is well-prepared.
As a non-Canadian, I would like to ask the following questions:
- line 38,39,40: more information on access to COVID-19 vaccines will be helpful (there were any public campaigns? how to homeless people encouraged to visit vaccination centers? was the health insurance mandatory to receive vaccine?)
- do the Authors collect data on the type of vaccine administered? Whether the single-dose (J&J) vaccine preferred, as only one visit was needed?
- how the other countries can learn from the Toronto experience? This will be interesting for the international community, as homelessness is going to be a significant public health/social policy challenge
Author Response
This study is scientifically correct. The aim of the study is clearly defined. The methodology section is well-prepared.
As a non-Canadian, I would like to ask the following questions:
- line 38,39,40: more information on access to COVID-19 vaccines will be helpful (there were any public campaigns? how to homeless people encouraged to visit vaccination centers? was the health insurance mandatory to receive vaccine?)
Response: Thank you for your thoughtful review of our study. We’ve modified the paragraph indicated to include more information about the setting in which COVID-19 vaccination occurred (and continues to occur) in Toronto, and have additionally added further context to the Methods section pertaining to the study’s setting (Setting and Design).
- do the Authors collect data on the type of vaccine administered? Whether the single-dose (J&J) vaccine preferred, as only one visit was needed?
Response: Unfortunately, the Johnson & Johnson vaccine was never used to a great extent in Ontario (or Canada), having been the last vaccine to receive Health Canada approval and shortly afterwards being held back due to concerns relating to side effects. That said, the COVENANT study did collect type of vaccine administered, and we now provide a table in the Supplement that describes the proportion of vaccinated participants by vaccine product.
- how the other countries can learn from the Toronto experience? This will be interesting for the international community, as homelessness is going to be a significant public health/social policy challenge
Response: As much as we would like to credit particular actions or strategies, our data does not explicitly link the high vaccination rate we found to any particular action or strategy implemented by parties responsible for the vaccination effort. However, as mentioned in our discussion, we do believe our results were, at least in part, driven by things like aggressive advocacy, widespread outreach efforts in collaboration with trusted community partners, and a general population setting where vaccination rates were very high. We’ve edited our discussion to more clearly emphasize this, and fully agree vaccination and other mitigation strategies for COVID-19 infection among people who are homeless will continue to be an important public health challenge.